# Circulating Neutrophils Do Not Predict Subclinical Coronary Artery Disease in Women with Former Preeclampsia

**DOI:** 10.3390/cells9020468

**Published:** 2020-02-18

**Authors:** John A.L. Meeuwsen, Judith de Vries, Gerbrand A. Zoet, Arie Franx, Bart C. J. M. Fauser, Angela H. E. M. Maas, Birgitta K. Velthuis, Yolande E. Appelman, Frank L. Visseren, Gerard Pasterkamp, Imo E. Hoefer, Bas B. van Rijn, Hester M. den Ruijter, Saskia C.A. de Jager

**Affiliations:** 1Laboratory for Experimental Cardiology, University Medical Center Utrecht, 3584 CX Utrecht, The Netherlandsjudithdevries222@hotmail.com (J.d.V.); H.M.denRuijter-2@umcutrecht.nl (H.M.d.R.); 2Wilhelmina Children’s Hospital Birth Center, University Medical Center Utrecht, 3584 CX Utrecht, The Netherlands; G.A.Zoet-2@umcutrecht.nl (G.A.Z.); A.Franx-2@umcutrecht.nl (A.F.); b.vanrijn@erasmusmc.nl (B.B.v.R.); 3Department of Reproductive Medicine and Gynaecology, University Medical Center Utrecht, 3584 CX Utrecht, The Netherlands; B.C.Fauser@umcutrecht.nl; 4Department cardiology, Radboud University Medical Center, 6525 GA Nijmegen, The Netherlands; angela.maas@radboudumc.nl; 5Department of Radiology, University Medical Center Utrecht, 3584 CX Utrecht, The Netherlands; B.K.Velthuis@umcutrecht.nl; 6Amsterdam University Medical Centre, VU Medical Centre, VU University, 1081 VV Amsterdam, The Netherlands; y.appelman@amsterdamumc.nl; 7Department of Vascular Medicine, University Medical Center Utrecht, 3584 CX Utrecht, The Netherlands; F.L.J.Visseren@umcutrecht.nl; 8Central Diagnostic Laboratory, University Medical Center Utrecht, Utrecht, 3584 CX Utrecht, The Netherlands; g.pasterkamp@umcutrecht.nl (G.P.); I.Hoefer@umcutrecht.nl (I.E.H.); 9Center Translational Immunology, University Medical Center Utrecht, Utrecht, 3584 CX Utrecht, The Netherlands

**Keywords:** coronary artery disease, preeclampsia, neutrophils, women

## Abstract

Introduction: Preeclampsia (PE) represents a hypertensive pregnancy disorder that is associated with increased cardiovascular disease (CVD) risk. This increased risk has been attributed to accelerated atherosclerosis, with inflammation being a major contributor. Neutrophils play an important role in the onset and progression of atherosclerosis and have been associated with vascular damage in the placenta as well as the chronic inflammatory state in women with PE. We therefore investigated whether circulating neutrophil numbers or reactivity were associated with the presence and severity of subclinical atherosclerosis in women with a history of PE. Methods: Women aged 45–60 years with a 10 to 20 years earlier history of early onset preeclampsia (delivery <34 weeks of gestation) (n = 90), but without symptomatic CVD burden were screened for the presence of subclinical coronary artery disease (CAD) using both contrast-enhanced and non-contrast coronary CT angiography. Subclinical CAD was defined as a coronary artery calcium (CAC) score ≥100 Agatston Units and/or ≥50% coronary luminal stenosis. We assessed whether the numbers and activity of circulating neutrophils were associated with the presence of subclinical CAD and as secondary outcome measurements, with the presence of any calcium (CAC score > 0 AU) or stenosis, categorized as absent (0%), minimal to mild (>0 and <50%), and moderate to severe (≥50%) narrowing of the coronary artery. Blood was drawn just before CT and neutrophil numbers were assessed by flow cytometry. In addition, the presence of the chemokine receptors CXCR2 and CXCR4, which are known to be instrumental in neutrophil recruitment, and neutrophil activity upon stimulation with the bacterial peptide N-Formylmethionyl-leucyl-phenylalanine (fMLF) was assessed by flow cytometry. Results: Of the participating women, with an average age of 49 years, 13% (12 out of 90) presented with subclinical signs of CAD (CAC score ≥100 AU and/or ≥50% luminal stenosis), and 37% (33 out of 90) had a positive CAC score (>0). Total white blood cell count and neutrophil counts were not associated with the presence of subclinical CAD or with a positive CAC score. When assessing the presence of the chemokine receptors CXCR4 and CXCR2, we observed a slight decrease of neutrophil CXCR2 expression in women with CAC (median MFI 22.0 [interquartile range (IQR) 20.2–23.8]) compared to women without CAC (23.8 [IQR 21.6–25.6], *p* = 0.02). We observed no differences regarding neutrophil CXCR4 expression. In addition, expression of the early activity marker CD35 was slightly lower on neutrophils of women with subclinical CAD (median MFI 1.6 [IQR 1.5–1.9] compared to 1.9 [IQR 1.7–2.1] in women without CAD, *p* = 0.02). However, for all findings, statistical significance disappeared after adjustment for multiple testing. Conclusion: Our findings indicate that neutrophil counts and (re)activity are not directly associated with silent CAD disease burden and as such are not suitable as biomarkers to predict the presence of subclinical CAD in a high-risk population of women with a history of preeclampsia.

## 1. Introduction

Cardiovascular disease (CVD) is a major health problem and annually causes 17.7 million deaths worldwide [1]. Women with a history of preeclampsia (PE), a hypertensive pregnancy disorder responsible for severe maternal and fetal morbidity and mortality [2,3], are at high risk of developing premature CVD, including coronary artery disease (CAD) [4,5,6,7]. Therefore, risk stratification is important to help prevent CAD in high-risk patients as recommended in clinical guidelines [8,9,10,11]. However, adequate biomarkers to identify women with former PE who are at high risk for CAD are currently not available. 

Atherosclerosis and preeclampsia (PE, defined as hypertension developing after 20 weeks of gestation in combination with either proteinuria, maternal organ dysfunction, or uteroplacental dysfunction) share common pathophysiologic mechanisms such as endothelial dysfunction and increased inflammation [12]. In PE patients, hypertension and neutrophil activation enhance the development of endothelial dysfunction [13]. Neutrophil activity was profoundly increased during pregnancy in women with PE as compared to women with normal pregnancy [14,15,16,17,18,19]. Moreover, high neutrophil numbers in the first trimester have been associated to early-onset preeclampsia and PE patients had increased levels of urinary neutrophil gelatinase-associated lipocalin (NGAL) [20,21].

The pathology of atherosclerosis is characterized by inflammation in the context of high lipids and reflected by the presence of circulating inflammatory cells [22,23,24,25,26]. Neutrophils play a major role in the development and rupture of atherosclerotic plaques [27,28,29]. Infiltrating neutrophils contribute to necrotic core formation [26] and induce plaque destabilization by the release of metalloproteinases and myeloid peroxidase [29,30]. In addition, neutrophils can enhance recruitment and local activation of monocytes [31], thereby increasing the necrotic core and enhancing plaque vulnerability [32]. Peripheral neutrophil counts have been significantly associated with lesion size and risk of recurrent cardiovascular events in patients [27,33,34,35,36], indicating that regulation of neutrophil homeostasis affects atherosclerosis [26]. Neutrophil migration is regulated by chemokine receptors CXCR2 and CXCR4. Increased CXCR2 and decreased CXCR4 expression is associated with enhanced neutrophil release from the bone marrow and impaired homing towards the bone marrow. Reduced expression of CXCR4, which increases neutrophil mobilization and reduces homing to the bone marrow, was associated with increased atherosclerotic lesion size in mice [37,38]. In line with these findings, CXCR4 expression on circulating neutrophils was lower in carotid endarterectomy patients and patients with unstable angina pectoris as compared to healthy controls [38].

Although an important role for neutrophils has been established in the pathology of both atherosclerosis and preeclampsia, little is known about the predictive value of neutrophils for subclinical CAD in women with former preeclampsia. In the current study, we investigated if neutrophil counts, neutrophil recruitment, and neutrophil activity were associated with subclinical CAD in women with a history of PE.

## 2. Methods

### 2.1. Participants

The current cross-sectional study comprises women from the CREW-IMAGO study [39,40]. We included women aged 45 to 60 years old with a history of early preeclampsia (delivery <34 weeks of gestation). Participants underwent a cardiovascular risk screening and coronary computed tomography (CCT), including contrast-induced CT, in the University Medical Center Utrecht. Preeclampsia was defined according to the 2014 definition of the International Society for the Study of Hypertension in Pregnancy as hypertension developing after 20 weeks of gestation in combination with either proteinuria, maternal organ dysfunction or uteroplacental dysfunction [41]. Women with any serious illness that compromised study participation were excluded, as well as patients with high risk for contrast nephropathy (renal dysfunction with an estimated glomerular filtration rate <60 ml/min/1.73 m^2^) or patients with a history of myocardial infarction. The study protocol conforms to the Declaration of Helsinki and has been approved by the local ethics committee on research on humans. All participants provided written informed consent upon inclusion.

### 2.2. Coronary Artery Calcium (CAC) and Plaque Measurement

The imaging methods for the visualization of calcification and stenosis are described in detail before [39] and analyzed by an experienced cardiovascular radiologist (BKV) using a standardized protocol. Briefly, a non-contrast CCT was performed to calculate the coronary artery calcium (CAC) score with the Agatston scoring method [42]. Presence of CAC was defined as >0 Agatston Units (AU). Next, coronary CT angiography was performed to assess plaque burden and luminal stenosis according to the American Heart Association classification [43,44]. A primary outcome was subclinical CAD, defined as CAC score ≥100 AU and/or ≥50% luminal stenosis. Secondary outcomes included the CAC score in Agatston Units, converted to MESA percentiles [45] and the presence of luminal stenosis was categorized as absent (0%), minimal to mild (>0 and <50%), and moderate to severe (≥50%) narrowing, based on diameter measurements comparing diameters of the maximal stenosis to a reference diameter proximal and distal to the stenotic area [46]. 

### 2.3. Blood Collection, Stimulation of Neutrophils and Red Blood Cell Lysis

Prior to the CT imaging procedure, 10 mL blood was collected in Natrium-Heparin anticoagulant tubes. A complete blood count profile was determined by a general hematology cell counter (Cell Dyn 1800 Abbott, MN, USA). We assessed both baseline activity of neutrophils, and neutrophil activity response upon whole blood stimulation. In order to stimulate the neutrophils, 2 mL of blood was incubated with the chemotactic peptide N-Formylmethionyl-leucyl-phenylalanine (fMLF, end concentration 1 µM, F3506, Sigma, St. Louis, MO, USA) for 5 minutes at 37 °C, within 10 minutes after blood draw. Activation was confirmed with flow cytometry analyses (described below) by increased expression of degranulation markers CD35, CD11B, and CD66B. As expected, CD62L surface receptors shed upon fMLF stimulation (*p* < 0.001 for all markers) (Figure 1A). To indicate the response of neutrophils upon stimulation with fMLF, we used the index (MFI after fMLF stimulation /MFI before stimulation) (Figure 1B).

After stimulation, red blood cells (RBCs) were lysed with RBC lysis buffer (deionized water supplemented with 155mM NH_4_Cl, 10mM KHCO_3_, and 0.1 mM EDTA, osmolality 305–310, and pH 7.4 at 4 °C) for 20 minutes on ice. To remove any remaining RBCs, the pellet was resuspended in RBC lysis buffer and centrifuged at 350 g for 5 min at 4 °C. To wash the cells, the pellet was resuspended and centrifuged at 350g for 5 min at 4 °C twice with PBS^2+^ (PBS supplemented with Albumin 10% *w*/*v* and trisodium citrate 0.32% *w*/*v*).

### 2.4. Flow Cytometry

To identify different neutrophil subsets and activity markers, the cells were incubated with antibodies (Appendix A) for 60 minutes at 4 °C protected from light. To remove any unbound antibodies, cells were washed with PBS. To distinguish dead cells, the cells were then incubated with fixable viability dye (eFluor 506, eBioscience, San Diego, CA, USA) for 30 min on ice, protected from light. After washing with PBS^2+^, the cells were measured by flow cytometry (Gallios, Beckman Coulter). Viable neutrophils were identified using forward and sideward scatter measurements in combination with markers for viability and CD9 (to exclude eosinophils). From the selected neutrophils, we measured the surface expression of chemokine receptors CXCR2 and CXCR4, markers of early (CD35), intermediate (CD11B), and late stage (CD66B) degranulation, and activity marker CD62L, which is shed upon activation. Both neutrophils from normal blood samples and activated neutrophils were measured by flow cytometry and analyzed with statistical methods as described below.

### 2.5. Statistical Analyses

Continuous variables were depicted as means and standard deviations or medians and interquartile ranges. Variables with normal distributions were compared using Student’s *t*-tests or one-way ANOVA and non-normal distributed variables were compared with the non-parametric Mann–Whitney *U* or Kruskal–Wallis tests. Categorical variables were analyzed with Chi-square or Fisher’s exact tests. Data management and statistical analyses were performed with RStudio [47] and the R software package (version 3.2.0. Vienna, Austria) [48]. Since several comparisons were performed, *p*-values were corrected for multiple testing using the false discovery rate (FDR) method [49]. We considered adjusted *p*-values <0.05 as statistically significant. Graphs were produced using the Graphpad Prism 7.02 software (La Jolla, CA, USA).

## 3. Results

### 3.1. Baseline Characteristics

The current study comprised 90 women with former preeclampsia, of which 12 (13%) had subclinical coronary artery disease (CAD), defined as a CAC score ≥100 Agatston Units (AU) and/or ≥50% luminal stenosis. Baseline characteristics were stratified by the presence of subclinical CAD (Table 1). Mean age was 49 years and did not differ between groups. Women with subclinical CAD tended to have a higher waist circumference (mean 94.6 ± 11.6 cm) compared to controls (88.4 ± 11.9 cm, *p* = 0.09). A positive family history of premature CVD was slightly more common in women with subclinical CAD (66.7%) than in women without CAD (39.7%), although this difference did not reach statistical significance (*p* = 0.15). The frequency of other CVD risk factors, such as hypertension, diabetes and metabolic syndrome was comparable between women with and without CAD. Of all participants, 57 (63%) had a CAC score of zero. The median CAC score in the group with CAC >0 AU was 35 AU (range 0.5–1989), and the median CAC percentile based on MESA classification was 93 (range 61–99) [45].

### 3.2. Total Neutrophil Numbers in Subclinical CAD

We first investigated the white blood cell (WBC) composition in our population. WBC counts and WBC subtypes were within the clinical reference values (data not shown). Similarly, in our patient population we did not observe any statistically significant differences in total WBC, lymphocyte and monocyte counts between women with and without CAC (defined as >0 AU), among women with different stenosis categories, or between women with and without subclinical CAD (Figure 2A–C). Although the granulocyte count was slightly increased in women with subclinical CAD (4.6 [IQR 3.6–5.7] G/L) as compared to women without CAD (4.1 [IQR 3.3–5.3] G/L), statistical significance was not reached (*p* = 0.45).

### 3.3. Neutrophil Chemokine Receptor Expression and Subclinical CAD

Next, we explored if neutrophil migration receptors CXCR2 and CXCR4 were associated with the presence of any coronary calcium, stenosis, or subclinical CAD. No differences were observed when the percentage of cells negative for CXCR2 or positive for CXCR4 were compared between women with or without CAC, stenosis, or subclinical CAD (Figure 3A–C). While the percentage of CXCR2 negative cells was not different, CXCR2 expression was slightly lower in women with CAC (median 22.0 [IQR 20.2–23.8] as compared to women without (median 23.8 [IQR 21.6–25.6], *p* = 0.02) (Figure 3D). However, after adjustment for multiple testing, the results were not statistically significant (*p* = 0.13). Neutrophil CXCR4 expression was not associated with CAC presence (Figure 3D). Surface expression of neutrophil migration markers CXCR2 and CXCR4 was not significantly associated with the presence of coronary stenosis or subclinical CAD (Figure 3E,F).

### 3.4. Neutrophil Activity and Subclinical CAD

Neutrophil activity was comparable between patients with and without CAC (Figure 4A). We then investigated whether neutrophil activity markers were associated with the degree of coronary stenosis in women with former PE. Upon fMLF stimulation, the response of neutrophils was lower for CD35 in PE patients with 50%–99% stenosis (1.5 [IQR 1.5–1.6]) as compared to PE patients without stenosis (1.9 [IQR 1.7–2.0], *p* = 0.03) (Figure 4B). However, after adjustment for multiple testing, the results were not statistically significant (*p* = 0.16). In line, the CD35 response was also lower in patients with subclinical CAD(1.6 [IQR 1.5–1.9]) than in patients without CAD (1.9 [IQR 1.7–2.1], *p* = 0.02), although the results were not statistically significant when multiple testing correction was applied (*p* = 0.11). A similar trend was observed for CD11B in women with coronary stenosis (Figure 4B) and with subclinical CAD (Figure 4C).

## 4. Discussion

According to the 2016 European guidelines on cardiovascular disease prevention, a periodic screening for hypertension and diabetes mellitus should be considered in women with a history of pre-eclampsia [11]. However, recent data from the CREW-IMAGO study showed that between 45 and 55 years of age, 30% of the women with a history of PE had an elevated CAC score as compared to 18% in the control group [40], demonstrating the importance of early identification of women with a history of PE that are at high risk for cardiovascular complications. Finding a biomarker to identify these women at high risk is crucial and such a marker may be found in players that are involved in both PE and CVD. Therefore, we investigated if women after PE with subclinical CAD had a different profile of neutrophil subsets or neutrophil (re)activity compared to women without evidence of subclinical CAD. Despite the well-established role of neutrophils in both PE and atherosclerosis, we found no differences in neutrophil numbers or (re)activity in cases compared to controls, except for a slight decrease in expression of the early neutrophil activity marker, CD35. These results suggest that neutrophil numbers or (re)activity are not useful to distinguish subclinical CAD cases from controls in women with a history of PE at middle-age.

Neutrophils have been shown to play an important role in the pathophysiology of PE [14,15,16,19]. As an easy biomarker available for routine clinical diagnostics the neutrophil to lymphocyte ratio has been tested as a predictive marker for PE on several occasions [50,51,52,53] and a recent. META analysis showed that NLR has an acceptable sensitivity as a diagnostic marker for PE [54]. Neutrophil to lymphocyte ratio at time of delivery did not correlate to pregnancy outcomes (i.e., birth weight, time from pre-eclampsia diagnosis to delivery and placental parameters (being weight and number of infarcts) [55], suggesting NLR mostly reflects the general inflammatory status in these women. In addition, it has been well-established that neutrophil counts are associated with the presence of CAD and the risk of recurrent CVD events [34,35,36,56,57]. Moreover, experimental studies have identified a crucial role for neutrophils in atherosclerotic development and progression [26,30,31]. Neutrophils contribute to the oxidation of lipoproteins and recruit monocytes to the atherosclerotic lesion [26] and neutrophil counts are correlated with atherosclerotic lesion area [27]. Mice lacking neutrophil granule protein cathelicidin (CRAMP or LL37 in human), an important activator and recruiter of immune cells, show reduced lesion size and less macrophage infiltration [31]. Recruited neutrophils from the bone marrow show low CXCR4 and high CXCR2 expression. Inhibition of CXCR2 led to impaired neutrophil recruitment in atherosclerotic lesions [27]. We found that women with coronary CAC tended to exhibit lower neutrophil CXCR2 expression, which might seem counterintuitive. However, since decreased neutrophil CXCR2 surface is associated with increased sequestration in the bloodstream and increased neutrophil activity [58], these results might suggest that activated neutrophils tend to remain in the bloodstream in women with CAC. Interestingly, rheumatoid arthritis patients, being at increased risk of CVD, also exhibited low neutrophil CXCR2 expression in the synovial fluid as compared to the peripheral blood, which might be the result of recruitment of neutrophils via IL-8 signaling or the local presence of TNF-α in the synovial fluid [59]. In addition, (non-neutrophil specific) CXCR2 expression was decreased in preeclamptic placentas [60]. Further research elaborating the pathways that involve CXCR2 signaling, for example the Akt pathway, might shine new light on the role of CXCR2 in PE patients that develop CAD.

We also found that upon stimulation with fMLF, neutrophil surface expression of degranulation marker CD35 (CR1) was marginally lower in women with subclinical CAD as compared to controls. These results might indicate that neutrophils become less activated in response to fMLF stimulation, which was also reflected by a trend of decreased neutrophil CD11B expression in formerly PE women with subclinical CAD. On the other hand, one may speculate that decreased expression of early degranulation marker CD35 indicates accelerated activation of neutrophils, but that should be accompanied with increased expression of the later degranulation markers CD11B and CD66B or by decreased CD62L expression, which we did not observe. In other studies, the role of CD35 remains to be fully elucidated. CD35 was higher in sepsis patients and also found in a soluble form in the synovial fluid of patients with rheumatoid arthritis [61,62]. Studies on CD35 polymorphisms in CVD patients have found contrasting results. In one study, CD35 polymorphisms were associated with increased inflammation and increased risk of MI [63]. In contrast, another study reported a higher frequency of a CD35 polymorphism in healthy controls than in CAD patients [64]. However, no functional evidence for neutrophil involvement was provided in these studies, as the authors linked the underlying biological mechanism mainly to CD35 on erythrocytes. 

The lack of differences in neutrophil (re)activity among women with former PE with or without CAD might have different explanations. First, it might be possible that neutrophils are activated in all women with former PE, but that these differences are not associated with presence of subclinical CAD. Although not the goal of the current study, a control population of women with a normal pregnancy might elaborate this hypothesis. Second, gender differences regarding neutrophil counts and activity have been reported in both experimental models and human studies [65,66,67,68,69], which may complicate the translation from previous atherosclerosis studies to the current study. For example, female mice had lower circulating neutrophil numbers [65] and reproductive female rats showed higher phagocytic response than males and pre- or post-reproductive females [67]. In humans, neutrophil extracellular traps, represented by MPO-DNA complexes were also found to be higher in males than females [68]. These differences might complicate the proper understanding of a possible role for neutrophils in women with former PE at high risk for CAD. Third, the available human evidence for neutrophil involvement in both PE and atherosclerosis was predominantly derived from acute manifestations of the diseases. Therefore, the involvement of neutrophils in PE might be temporal and not present during the subclinical phase. Indeed, when neutrophil activity was assessed in PE patients, CD11B expression was higher in PE patients compared to controls, but the differences disappeared six weeks and six months postpartum [19]. In line, neutrophil superoxide production was increased in PE patients as compared to controls during pregnancy but disappeared after delivery. In a control group of pregnant women with essential hypertension, superoxide production was also increased, and remained high postpartum, indicating that neutrophil activity might rather result from hypertension, than from PE [16]. In the current study, we could not test this hypothesis since we could not expose healthy individuals to a dose of 3.0 mSv for research purposes, because of regulatory and ethical restrictions. When comparing former PE women with and without hypertension, however, neutrophil count, migratory capacity and activity did not differ (data not shown).

## 5. Conclusions

We performed our research on a unique population of women with a history of PE in which we assessed CAD by contrast-enhanced and non-contrast coronary CT angiography. Due to ethical considerations regarding the radioactive contrast imaging performed in this study we were not able to include a control population of women without PE. Furthermore, our cohort is relatively small, with only a limited subgroup of women with evident CAD. This may have impacted the outcome of our study and to firmly establish our finding these experiments should be validated in a larger cohort. Lastly, the current study design did not allow us to associate neutrophil numbers or their activation state to characteristics of plaque morphology and specifically plaque instability.

To conclude, the current study showed no predictive value at cross-sectional level for subclinical CAD and therefore long-term follow-up to appreciate the prognostic value of neutrophil counts and neutrophil activity for the occurrence of cardiovascular events in these women at high risk remains to be established. Distinguishing women at high risk for CVD remains crucial. However repetitive invasive CT scans of all women with a history of PE may not be favorable and as such the need for a biomarker remains evident. Other inflammatory cells or markers that are involved in the pathology of both PE and atherosclerosis [12] or sex-specific markers involved in atherosclerosis such as growth differentiation factor 15 [70] may be of specific interest. 

Thus, although neutrophils are important in both atherosclerosis and preeclampsia, these findings indicate that neutrophil counts and (re)activity are not directly associated to subclinical CAD disease burden and as such are not suitable as biomarkers to predict the presence of subclinical CAD in women with a history of preeclampsia.

## Figures and Tables

**Figure 1 cells-09-00468-f001:**
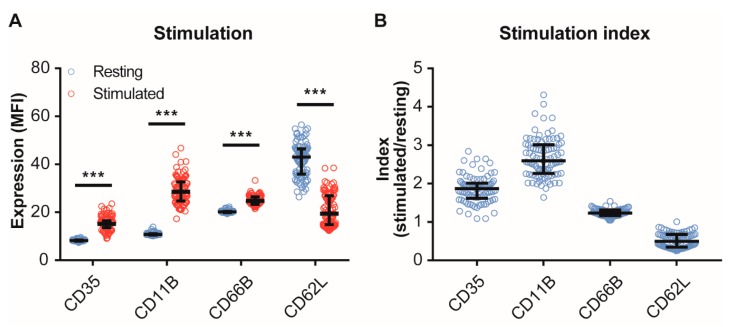
Surface expression of neutrophil activity markers. Upon stimulation with N-Formylmethionyl-leucyl-phenylalanine (fMLF, 1 μM), surface expression of activity markers CD35, CD11B, and CD66B increases, whereas CD62L surface expression decreases (**A**). Moreover, the index values (after/before stimulation with fMLF) are shown (**B**). Data are presented as median with interquartile ranges from all patients (n = 90). *** indicates *p* < 0.001 (Mann–Whitney U test).

**Figure 2 cells-09-00468-f002:**
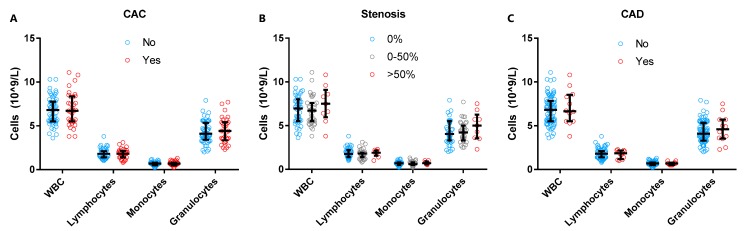
White blood cell (WBC), lymphocyte, monocyte, and neutrophil counts of the total study population stratified by presence of coronary artery calcification (CAC) (**A**), stenosis (**B**), and subclinical coronary artery disease (CAD) (**C**). Although the granulocyte count is slightly increased in former PE women with 50%–99% stenosis and subclinical CAD, these differences were not statistically significant. CAC indicates coronary artery calcification defined as >0 Agatston Units (AU); CAD, subclinical coronary artery disease defined as ≥100 AU and/or ≥50% stenosis.

**Figure 3 cells-09-00468-f003:**
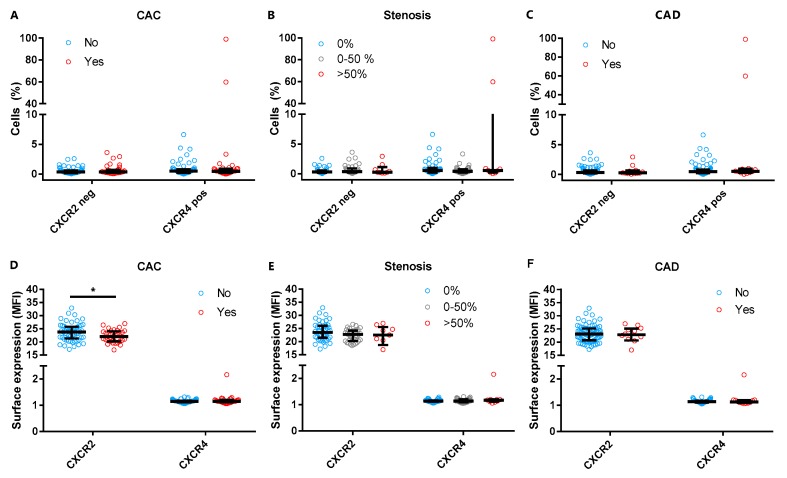
Presence of chemokine receptors CXCR2 and CXCR4 is shown in women with a history of PE. The percentages of cells expressing CXCR2 or CXCR4 were comparable between women with or without presence of CAC, stenosis, and subclinical CAD (**A**–**C**). While the percentage of CXCR2 negative cells was comparable (**A**), CXCR2 surface expression was lower in women with CAC as compared to women without CAC (**D**). We observed no differences for CXCR2 or CXCR4 regarding the presence of stenosis or subclinical CAD (**E**,**F**). CAC indicates coronary artery calcification defined as >0 Agatston Units (AU); CAD, subclinical coronary artery disease defined as ≥100 AU and/or ≥50% stenosis. * indicates *p* < 0.05 (Mann–Whitney *U* test).

**Figure 4 cells-09-00468-f004:**
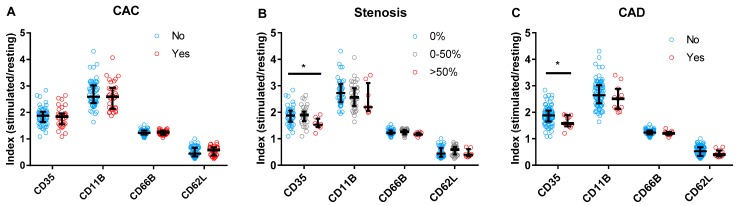
Neutrophil activity is shown in women with former PE. Neutrophil activity was not different between women with and without CAC (**A**). In women with >50% of coronary stenosis and with subclinical CAD, the neutrophil CD35 index was lower, indicating a lower response to stimulation with the chemotactic peptide fMLF (**B**,**C**). A similar trend was observed for degranulation marker CD11B, but not for CD66B and CD62L (**B**,**C**). CAC indicates coronary artery calcification defined as >0 Agatston Units (AU); CAD, subclinical coronary artery disease defined as ≥100 AU and/or ≥ 50% stenosis. * indicates *p* < 0.05 (Kruskal–Wallis test and Mann–Whitney U test).

**Table 1 cells-09-00468-t001:** Baseline characteristics of subclinical coronary artery disease (CAD) cases and controls.

	Controls	CAD Cases	*p*
	n = 78	n = 12	
**Patient characteristics**			
Age (years)	49.4 ± 3.7	49.0 ± 5.0	0.72
GA delivery (days)	213.0 ± 28.4	205.6 ± 20.4	0.39
**Clinical measurements**			
Systolic blood pressure (mmHg)	130.8 ± 16.0	127.5 ± 15.1	0.50
Diastolic blood pressure (mmHg)	80.1 ± 10.0	77.7 ± 10.6	0.43
BMI (kg/m2)	27.6 ± 5.1	29.7 ± 6.9	0.20
Waist circumference (cm)	88.4 ± 11.9	94.6 ± 11.6	0.09
Total cholesterol (mmol/L)	5.3 [4.8–6.0]	5.6 [4.8–5.8]	0.87
Triglycerides (mmol/L)	1.1 [0.8–1.5]	1.2 [0.9–1.5]	0.54
HDL-cholesterol (mmol/L)	1.5 [1.3–1.6]	1.5 [1.3–1.6]	1.00
LDL-cholesterol (mmol/L)	3.3 [2.8–3.9]	3.4 [2.7–3.8]	0.64
Glucose (mmol/L)	5.5 ± 1.3	5.3 ± 0.5	0.57
**CVD risk factors**			
Family history of premature CVD (no, %)	31 (39.7)	8 (66.7)	0.15
Hypertension^a^ (no, %)	42 (54.5)	8 (66.7)	0.64
Obesity (no, %)	23 (29.5)	4 (33.3)	0.75^†^
Diabetes (no, %)	3 (3.8)	0 (0.0)	1.00^†^
Current smoking (no, %)	8 (10.5)	0 (0.0)	0.59^†^
Metabolic syndrome^b^ (no, %)	25 (32.1)	4 (33.3)	1.00^†^
FRS (percentage)	6.4 ± 4.0	5.5 ± 3.5	0.48
Intermediate–high risk, FRS ≥10% (no, %)	12 (15.8)	1 (9.1)	1.00^†^
**Whole blood count**			
WBC (G/L)	6.8 [5.5–7.8]	6.6 [5.5–8.4]	0.79
Lymphocytes (G/L)	1.8 [1.4–2.1]	1.8 [1.3–2.1]	0.97
Monocytes (G/L)	0.7 [0.5–0.8]	0.7 [0.6–0.7]	0.72
Granulocytes (G/L)	4.1 [3.3–5.3]	4.6 [3.6–5.7]	0.45

The demographic characteristics are stratified by presence of coronary artery disease. The values are presented as mean ± standard deviation for normal distributions, number of patients (frequency in percentage) for categorical variables, and median [interquartile range] for non-normal distributions. *P*-values are calculated using the Student’s *t*-test, Chi-square or Fisher’s exact test (†), and Mann–Whitney *U* test, respectively. CAD indicates subclinical coronary artery disease defined as ≥100 Agatston Units and/or ≥50% stenosis; GA, gestational age; BMI, body-mass index; HDL, high-density lipoprotein; LDL, low-density lipoprotein; FRS, Framingham Risk Score; WBC, white blood cells. ^a^ Blood pressure ≥140/90 mmHg or current use of antihypertensive treatment. ^b^ According to NCEP ATP-III criteria.

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
