# Peer review of "Circulating Neutrophils Do Not Predict Subclinical Coronary Artery Disease in Women with Former Preeclampsia"

_cells, 2020, doi:10.3390/cells9020468_

Round 1

Reviewer 1 Report

Meeuwsen et. al. have investigated if neutrophils are a good candidate for prediction of coronary artery disease (CAD) in women with preeclampsia. The authors have used biochemical and biophysical measurements to determine possible associations of neutrophil counts with CAD. They have also investigated chemokine receptor expression under these conditions. Results show that the neutrophil counts are not directly correlated with CAD and hence, are not suitable biomarkers.

Study is conducted well and the results are communicated clearly. I will only suggest that the authors move the supporting figure 1 to the main text since it is an integral part of the study.

Author Response

We thank the reviewer for the positive evaluation of our manuscript. As suggested we have now moved the supplemental figure 1 to the main text.

Reviewer 2 Report

Specific comments:

Introduction is too long. Some parts may be move to discussion section. Study group is relatively small with strong imbalance between subgroups what may influence the results. Was the sample size calculated? Please add clear “Limitations” section.

Author Response

We thank the reviewer for the positive evaluation of our manuscript. Please find our point to point response to your suggestions below.

1. Introduction is too long. Some parts may be move to discussion section.

We have shortened the introduction and move some parts to the discussion as suggested by the reviewer.

2. Study group is relatively small with strong imbalance between subgroups what may influence the results. Was the sample size calculated?

We acknowledge that the cohort is relatively small, however considering the population of unique patients (Women aged 45-60 years with a 10 to 20 years earlier history of early onset preeclampsia) combined with the study design (contrast-enhanced and non-contrast coronary CT angiography and availability of fresh blood for neutrophil analysis) we believe the cohort is of reasonable size. As we set-out this experiments without prior knowledge on the degree and severity of CAD in these women we were not able to perform a reliable power calculation a priori. We have touched upon your concern regarding subgroup imbalance in the limitations section.

3. Please add clear “Limitations” section. 

We thank the reviewer for this suggestion. We have added a clear and concise limitation section, which is pasted below for the reviewers convenience.

Limitations to the study:

We performed our research on a unique population of women with a history of PE in which we assessed CAD by contrast-enhanced and non-contrast coronary CT angiography. Due to ethical considerations regarding the radioactive contrast imaging performed in this study we were not able to include a control population of women without PE. Furthermore, our cohort is relatively small, with only a limited subgroup of women with evident CAD. This may have impacted the outcome of our study and to firmly establish our finding these experiments should be validated in a larger cohort. Lastly, the current study design did not allow us to associate neutrophil numbers or their activation state to characteristics of plaque morphology and specifically plaque instability.